# From Survival to Parenthood: The Fertility Journey After Childhood Cancer

**DOI:** 10.3390/biomedicines13081859

**Published:** 2025-07-30

**Authors:** Sofia Rahman, Veronica Sesenna, Diana Osorio Arce, Erika Maugeri, Susanna Esposito

**Affiliations:** Pediatric Clinic, Department of Medicine and Surgery, University of Parma, 43126 Parma, Italy; sofia.rahman@unipr.it (S.R.); veronica.sesenna@unipr.it (V.S.); diana.osorioarce@unipr.it (D.O.A.); erika.maugeri@unipr.it (E.M.)

**Keywords:** fertility preservation, childhood cancer survivors, gonadotoxicity, premature ovarian insufficiency, spermatogenesis, reproductive health

## Abstract

**Background**: The advances in cancer diagnosis and treatment have significantly improved survival rates in pediatric patients, with five-year survival now exceeding 80% in many high-income countries. However, these life-saving therapies often carry long-term consequences, including impaired fertility. The reproductive health of childhood cancer survivors has emerged as a key issue in survivorship care. **Objective**: This narrative review aims to examine the gonadotoxic effects of cancer treatments on pediatric patients, evaluate fertility preservation strategies in both males and females, and provide guidance on the long-term monitoring of reproductive function post treatment. **Methods**: A comprehensive literature review was conducted using PubMed, including randomized trials, cohort studies, and clinical guidelines published up to March 2024. The keywords focused on pediatric oncology, fertility, and reproductive endocrinology. Studies were selected based on relevance to treatment-related gonadotoxicity, fertility preservation options, and follow-up care. **Results**: Radiotherapy and alkylating agents pose the highest risk to fertility. Postpubertal patients have access to standardized preservation techniques, while prepubertal options remain experimental. Long-term effects include premature ovarian insufficiency, azoospermia, hypogonadism, and uterine dysfunction. The psychosocial impacts, especially in female survivors, are profound and often overlooked. **Conclusions**: Fertility preservation should be discussed at diagnosis and integrated into treatment planning in pediatric patients with cancer. While options for postpubertal patients are established, more research is needed to validate safe and effective strategies for younger populations. A multidisciplinary approach and long-term surveillance are essential for safeguarding future reproductive potential in childhood cancer survivors.

## 1. Introduction

Over the past few decades, the five-year survival rate for pediatric cancer patients has surpassed 80% in most European and North American countries, primarily due to the development of multimodal therapies such as chemotherapy, radiotherapy, and immunotherapy [1]. Parallel advancements in diagnostic tools and therapeutic strategies have further improved outcomes—enhancing survival in patients with previously poor prognoses, while allowing for the de-escalation of treatment intensity in those with more favorable outlooks [2].

However, prolonged survival brings new challenges. It is projected that by 2040, 73% of childhood cancer survivors will develop at least one chronic health condition, and 42% will suffer from a serious, life-threatening, or disabling illness—or die from a chronic disorder [3]. As survival improves, the long-term quality of life—including the ability to have biological children—has emerged as a major concern in pediatric oncology.

Infertility is one of the most impactful long-term side effects of cancer treatment in children. The degree of gonadal damage varies widely and depends on several factors, such as treatment type and intensity, the use of radiotherapy near reproductive organs, and the patient’s age and baseline gonadal function [4,5]. Alkylating agents, total body irradiation, and pelvic radiotherapy are among the most gonadotoxic treatments, while platinum-based drugs and anthracyclines carry an intermediate risk [6,7].

In male patients, cancer therapies can impair spermatogenesis or disrupt sperm transport [8]. In female patients, the risks include premature ovarian insufficiency, follicular depletion, and uterine damage, which may hinder the ability to conceive or carry a pregnancy [9]. Furthermore, damage to the hypothalamic–pituitary–gonadal axis from cranial irradiation or neurosurgery can interfere with the hormonal regulation essential for reproductive function [10].

Despite these risks, fertility preservation is often under-discussed or initiated too late. While standardized options exist for postpubertal patients, approaches for prepubertal children remain largely experimental and ethically complex. Additionally, oncologists may lack the resources or guidelines to navigate these conversations effectively [11]. This narrative review was conducted to address this critical gap. By summarizing current evidence on the gonadotoxic effects of pediatric cancer therapies and evaluating available fertility preservation strategies, we aim to provide clinicians with a comprehensive and practical resource. The review emphasizes sex-specific risks and methods, covering both prepubertal and postpubertal patients, and includes guidance on post-treatment fertility surveillance. In doing so, we hope to support earlier, more informed decision-making between healthcare providers, patients, and families—and ultimately, to help safeguard the reproductive futures of childhood cancer survivors. Although numerous reviews have addressed fertility issues in childhood cancer survivors, significant gaps remain regarding sex-specific risks, experimental options for prepubertal patients, and the psychosocial consequences of fertility loss. Moreover, recent advances in reproductive technologies and updated clinical guidelines necessitate a comprehensive synthesis of current evidence. This review aims to fill this gap by providing an up-to-date, detailed, and practical resource for clinicians managing pediatric patients facing fertility risks.

## 2. Materials and Methods

A comprehensive narrative literature review was conducted to evaluate the impact of cancer therapies on fertility, assess fertility preservation techniques in pediatric patients of both sexes, and identify optimal strategies for reproductive function monitoring after treatment. The search was performed using the PubMed database, covering articles published in English up to March 2024.

We used a combination of Medical Subject Headings (MeSH) and free-text keywords related to pediatric oncology, fertility, and reproductive health. The key search terms included “childhood cancer”, “pediatric oncology”, “infertility”, “fertility preservation”, “gonadotoxicity”, “cryopreservation”, “oocyte”, “sperm”, “ovarian reserve”, “testicular function”, “FSH”, “LH”, “AMH”, and “hypogonadism”. These were cross-referenced with oncological treatment terms such as “chemotherapy”, “radiotherapy”, “hematopoietic stem cell transplantation”, “surgery”, and “alkylating agents”.

Overall, 312 articles were initially screened based on the title and abstract, of which 97 met the inclusion criteria and were included in this review. No formal quality assessment tools (e.g., RoB tools) were applied given the narrative nature and broad scope of the review, but studies were evaluated for relevance, population characteristics, and data completeness.

The selection process prioritized studies focusing on pediatric populations (including adolescents and young adults up to age 21), with an emphasis on data specific to prepubertal and postpubertal patients. We included randomized controlled trials, systematic reviews, meta-analyses, prospective and retrospective cohort studies, case–control studies, interventional trials, and expert consensus guidelines.

Studies were included if they met the following criteria:Discussed the gonadotoxic effects of cancer therapies in pediatric patients;Described or evaluated fertility preservation methods applicable to children and adolescents;Provided data or recommendations on post-treatment fertility surveillance and reproductive health outcomes.

The references cited in relevant articles were also manually reviewed to identify additional sources of interest (snowball strategy). Special attention was given to recent clinical guidelines from reputable bodies such as the American Society of Clinical Oncology (ASCO) and European Society for Medical Oncology (ESMO), along with position statements from pediatric endocrine and reproductive societies.

This review synthesizes current knowledge and clinical practices to support healthcare professionals in addressing fertility-related issues in pediatric oncology patients—an increasingly critical component of survivorship care. In addition to narrative synthesis, we extracted and reported numerical data such as prevalence rates, incidence of fertility complications, cumulative chemotherapy doses, radiation thresholds, and relevant risk estimates to enhance the quantitative understanding of fertility outcomes in pediatric cancer survivors.

## 3. Fertility Preservation

### 3.1. Pediatric Female Patients

ASCO recommends that fertility preservation be discussed with the parents or guardians of all pediatric and adolescent patients at risk of infertility prior to the initiation of cancer therapy [12]. With 5-year survival rates for childhood cancers now exceeding 80%, the number of female survivors facing gonadal dysfunction due to chemotherapy or radiotherapy has significantly increased [13]. The estimated overall risk of infertility in girls following cancer treatment is approximately 16% [14].

As oncologic treatments become increasingly effective but also more intensive, there is a growing need to prioritize long-term quality of life—including reproductive health—during the early phases of therapeutic planning [15]. In this context, timely referral to a pediatric endocrinologist is essential to ensure appropriate evaluation and counseling from the moment of diagnosis [16].

Several factors must be considered when determining fertility preservation strategies in female pediatric patients, including the type and timing of cancer treatment, patient age, and pubertal status. While many preservation technologies were originally developed for adults, pediatric-specific protocols remain limited and under development [17,18].

A significant divergence exists between the management of prepubertal and postpubertal girls, influenced by disease stage and treatment urgency. This often leads to the abandonment of preservation techniques deemed too invasive or risky for younger patients [19]. For postpubertal girls, if cancer treatment can be delayed for up to two weeks, options include oocyte or embryo cryopreservation, which are considered standard methods. When a delay is not possible, ovarian tissue cryopreservation—although still experimental—is often the only feasible option [20].

Among standard techniques, embryo cryopreservation is the most established and offers high success rates. However, oocyte cryopreservation is often preferred by adolescents and young adults who do not have a long-term partner and may be reluctant to use donor sperm due to personal or ethical concerns [21]. Ovarian stimulation protocols must be carefully adapted for cancer patients, considering the limited time frame and the risk of temporarily elevated estradiol levels. The GnRH agonist protocol is typically preferred due to its shorter duration and safer hormonal profile [18].

In oocyte cryopreservation, mature eggs are retrieved and frozen, while embryo cryopreservation involves the fertilization of the oocyte with sperm prior to freezing. Both can later be thawed and used for assisted reproduction [22]. Notably, unused cryopreserved oocytes generally pose fewer ethical or emotional concerns for families than embryos, particularly in the event of patient death or disuse [23].

Technological advances in cryoprotectants, vitrification techniques, and intracytoplasmic sperm injection (ICSI) have substantially improved the outcomes of oocyte cryopreservation in recent years [24].

The role of gonadotropin-releasing hormone agonists (GnRH-a) during chemotherapy remains controversial. While some data suggest a potential protective effect on ovarian function, a consensus is lacking. For prepubertal girls, fertility preservation remains especially challenging. Due to the absence of mature oocytes and limited long-term outcome data, ovarian tissue cryopreservation is currently the only viable—yet still experimental—option [14]. Experience with hormone and fertility restoration in patients under two years of age remains scarce [18].

Ovarian tissue cryopreservation involves the laparoscopic removal of the ovarian cortex, where strips rich in primordial follicles are harvested for storage [13]. Two main cryopreservation techniques are used: slow freezing and vitrification [20]. Since the first documented live births from cryopreserved ovarian tissue in 2004 and 2005, the field has made substantial progress. As a result, ovarian tissue cryopreservation is increasingly recognized as a legitimate method for fertility preservation in select patients [18].

Finally, it is important to acknowledge the socioeconomic barriers to access. While many families can afford the initial procedures, the long-term cost of tissue storage and future use may be prohibitive—raising significant issues of equity and access in fertility care for pediatric oncology patients.

Figure 1 shows the main fertility preservation options and considerations for pediatric female cancer patients, highlighting differences based on pubertal status, available techniques, medical interventions, and access challenges.

### 3.2. Pediatric Male Patients

Male fertility depends on two essential functions of the testes: the production of testosterone and the generation of sperm. These processes are governed by the hypothalamic–pituitary–testicular (HPT) axis, which is activated during puberty. The increase in gonadotropin-releasing hormone (GnRH) stimulates luteinizing hormone (LH) secretion, which in turn drives testosterone production and initiates spermatogenesis. This complex process takes place within the seminiferous tubules and involves several key cell types: Sertoli cells, which support and nourish developing sperm; Leydig cells, which produce testosterone; and peritubular myoid cells, which contribute to the structural and functional integrity of the tubules [25].

Among these components, the germinal epithelium—responsible for sperm production—is particularly vulnerable to the cytotoxic effects of chemotherapy and radiotherapy. As a result, survivors of childhood cancer (CCSs) are at considerable risk of developing azoospermia (absence of sperm) or oligospermia (low sperm count) [26].

For postpubertal male patients, the ASCO guidelines recommend established fertility preservation methods similar to those used in adults. Sperm cryopreservation (sperm banking) remains the gold standard and should be routinely discussed with all postpubertal males undergoing gonadotoxic treatment. In contrast, GnRH agonists have not demonstrated efficacy in preserving male fertility and are not recommended [12].

Sperm cryopreservation is straightforward and can be performed at a fertility clinic, during hospitalization, or even at home, with proper arrangements for sample transport. Multiple semen samples are often needed to ensure an adequate reserve for future reproductive use [27]. However, this process can be especially challenging for adolescents and young adults (AYAs). Some patients may not have prior experience with masturbation, making the topic sensitive. Clinicians should approach the conversation with empathy and offer the patient the choice of involving their parents—balancing the need for privacy with the potential support parents can provide [28].

Despite the availability of these techniques, significant barriers often hinder sperm procurement among adolescent boys. Beyond medical eligibility, significant procedural and psychological barriers hinder sperm procurement in adolescent boys. Embarrassment, a lack of privacy, limited sexual experience, and anxiety about cancer diagnosis often discourage adolescents from providing a semen sample. Clinicians should approach fertility discussions sensitively, involve adolescent psychologists where needed, and offer alternative methods such as electroejaculation or testicular sperm extraction for patients unable or unwilling to provide samples via masturbation.

In cases where the patient is too ill or unable to produce a sample through masturbation, alternative collection methods may be employed. These include penile vibratory stimulation (PVS), rectal electroejaculation (EEJ), epididymal sperm aspiration, and testicular sperm extraction—though the latter options are more invasive and require specialized expertise [27].

The greatest challenge that remains is fertility preservation in prepubertal boys, as they cannot produce mature sperm. Currently, no clinically approved methods exist for this group. The most promising approach is testicular tissue cryopreservation, which is still considered experimental [12].

Recent advancements in experimental reproductive science have focused on the cryopreservation of spermatogonial stem cells (SSCs), in vitro spermatogenesis, and immature testicular tissue (ITT) cryopreservation followed by transplantation [29,30]. A landmark study in 2019 reported a successful birth in a rhesus macaque following the autologous transplantation of cryopreserved ITT, raising hope for future clinical applications in humans [31].

Cryopreservation techniques for ITT include controlled slow freezing (CSF) and vitrification, while transplantation strategies fall into two categories: autologous (within the same individual) and xenologous (between species) [32]. In autologous transplantation, factors such as graft size and implantation site significantly affect outcomes. Larger tissue fragments (>1 mm^3^) demonstrate better survival due to paracrine signaling that mitigates ischemic damage, while smaller fragments are more prone to resorption. Common transplantation sites studied include the back, shoulder, arm, and scrotal skin, with the latter showing the most promising results due to temperature stability and proximity to natural anatomical conditions [33,34,35,36].

Xenotransplantation, while not intended for clinical use due to risks such as immune rejection and zoonotic transmission, remains a valuable research tool. It has enabled complete spermatogenesis in several animal models and offers a unique window into testicular development across species [32].

A recent European study by Masliukaite et al. demonstrated that even prior to therapy, prepubertal cancer patients have significantly fewer spermatogonial cells compared with healthy controls (48.5% vs. 31.0%) [37]. The most affected were patients with central nervous system (CNS) and hematologic malignancies, likely due to tumor-related endocrine disruptions or systemic inflammation. For example, posterior fossa tumors may impact the hypothalamic–pituitary axis, while cytokine storms and persistent fever—common in leukemia and lymphoma—can impair testicular function.

Importantly, the timing of fertility preservation is critical. At diagnosis, families are often overwhelmed by the emotional burden of cancer, and discussions about future fertility may feel secondary. Nevertheless, fertility preservation should be introduced as an integral part of pre-treatment counseling, alongside other treatment-related decisions [38].

The effectiveness of this counseling depends greatly on the involvement of a specialized fertility team. Multidisciplinary collaboration ensures that families receive accurate, age-appropriate information and understand the risks, benefits, and limitations—particularly when considering experimental techniques. As some interventions may not result in successful fertility restoration, transparent and compassionate communication is essential [39,40].

Looking ahead, TTC holds significant potential for future clinical application. Advances in tissue grafting, SSC culture, and in vitro spermatogenesis may eventually enable fertility restoration in prepubertal boys. While complete in vitro spermatogenesis has been achieved in several animal models, translation into human applications remains in early stages due to challenges in replicating the complex testicular microenvironment. Nevertheless, these studies hold promise for restoring fertility in prepubertal boys unable to produce mature sperm at diagnosis.

Figure 2 summarizes the fertility preservation approaches in pediatric male cancer patients, distinguishing the available strategies for prepubertal and postpubertal boys, and addressing key clinical considerations, medical limitations, and access challenges.

## 4. Fertility After Cancer Treatment

### 4.1. Pediatric Female Patients

Females are born with a finite number of oocytes, as there is no postnatal proliferation of germ cells [41]. From fetal development onward, oocytes undergo progressive and irreversible atresia. At birth, the estimated number of oocytes is approximately 1.2 million, which declines to around 300,000 by puberty. Of these, only 300–500 will mature and ovulate throughout a woman’s reproductive lifespan, while the remainder undergo atresia.

It is important to distinguish the terms used to describe ovarian dysfunction, as their misuse can lead to confusion in both clinical practice and research. According to the 2016 ESHRE guidelines [42]:-“Acute ovarian failure (AOF)” refers to the immediate and complete loss of ovarian function occurring during or shortly after cancer treatment. It is characterized by amenorrhea and elevated gonadotropins soon after therapy initiation.-“Premature ovarian insufficiency (POI)” is defined as the cessation of ovarian activity before the age of 40, with amenorrhea lasting at least four months and elevated follicle-stimulating hormone (FSH) levels on two separate occasions. Unlike AOF, ovarian activity in POI can be intermittent and may allow spontaneous ovulation and even pregnancy.-“Premature menopause” is often used interchangeably with POI in common language but technically denotes the permanent and irreversible cessation of ovarian function before age 40, leading to definitive infertility and hypoestrogenism. While all premature menopause is POI, not all POI necessarily progresses to premature menopause.

The clear use of these definitions is essential for the consistent reporting of fertility outcomes and patient counseling.

POI affects approximately 0.9% of the general population; however, female survivors of childhood or adolescent cancer are at a significantly increased risk, with an estimated incidence of 8% by age 40 [43].

Among cancer survivors, two distinct forms of ovarian dysfunction are recognized: AOF and premature menopause [44]. Importantly, cancer treatments may also compromise uterine integrity. Pelvic or total body irradiation, for instance, can impair uterine structure and function, with potential consequences for future fertility and pregnancy outcomes [45].

Radiation therapy poses a dual threat to female reproductive health by affecting both ovarian and uterine tissues [46]. Although oocytes are arrested in prophase I and considered relatively inactive, they are paradoxically highly radiosensitive. While it was historically believed that oocytes lack the capacity to repair DNA damage, recent studies in mammalian models demonstrate that their DNA repair ability depends on developmental stage and radiation dose exposure [47,48,49].

In a retrospective study of 2196 female childhood cancer survivors, it was found that direct ovarian irradiation at doses ≥1000 cGy was associated with the highest risk of POI. Even lower doses (1–99 cGy) were linked to an increased risk compared with non-irradiated controls [10]. This confirms the exceptional radiosensitivity of ovarian tissue [47,50]. Anderson et al. further quantified age-specific threshold doses for POI of 20.3 Gy for infants, 18.4 Gy for children ≤10 years, and 16.5 Gy for adolescents ≤20 years, highlighting a negative correlation between patient age and radiation tolerance [51].

In addition to ovarian effects, radiation-induced uterine damage can affect the myometrium, endometrium, and vascular supply, increasing the risks of miscarriage, fetal growth restriction, perinatal mortality, preterm birth, small-for-gestational-age infants, preeclampsia, and abnormal placentation [52,53]. Even with preserved ovarian function, survivors often experience clinical infertility or delayed conception due to uterine sequelae [54].

Cranial irradiation, much like in males, may disrupt the hypothalamic–pituitary–gonadal (HPG) axis in females, leading to hypoestrogenism and amenorrhea. Furthermore, it has been associated with an increased miscarriage rate [55,56].

Chemotherapeutic agents—especially alkylating agents—also pose a significant risk to the ovarian reserve, as demonstrated across numerous studies [57,58]. High-dose cyclophosphamide, alone or with busulfan, is frequently used in preparative regimens for hematopoietic stem cell transplantation (HSCT) and is particularly gonadotoxic [59]. In contrast, agents such as taxanes and anthracyclines have moderate gonadotoxicity, while antimetabolites (e.g., methotrexate) and vinca alkaloids are considered minimally gonadotoxic [59]. Although a definitive threshold dose for alkylating agents has not been established, higher cumulative exposures significantly elevate POI risk [57].

Chemaitilly et al. identified cyclophosphamide and procarbazine as key contributors to AOF. Notably, procarbazine was found to induce AOF regardless of patient age, while cyclophosphamide posed a higher risk in postpubertal individuals [60]. A more recent study by Bjornard et al. emphasized the heightened gonadotoxicity of treatment regimens for solid tumors—such as neuroblastomas and sarcomas—where average cyclophosphamide equivalent doses (CEDs) reached 20.9 g/m^2^, compared with 4 g/m^2^ in leukemia/lymphoma protocols [61].

Unlike radiotherapy, most chemotherapeutic agents do not impair uterine function. However, previous exposure to doxorubicin or daunorubicin has been associated with an increased risk of delivering low-birth-weight infants [56].

With the advent of molecularly targeted therapies, the reproductive impact of newer agents remains poorly understood. However, preliminary data suggest that bevacizumab, a VEGF inhibitor, may transiently disrupt ovarian function by impairing angiogenesis—a critical process for follicular development [62].

In female patients undergoing abdominal or pelvic radiotherapy and/or chemotherapy, the annual monitoring of pubertal progression using Tanner staging is recommended until full sexual maturity is achieved. Hormonal assays—including FSH, LH, and estradiol—are essential to evaluate for hypogonadism and may inform the need for endocrine follow-up. After puberty, the ovarian reserve should be assessed using transvaginal ultrasound to measure antral follicle count (AFC), along with hormone levels (FSH, LH, estradiol, and AMH) to gauge reproductive potential [63,64,65].

The psychosocial burden of fertility loss is profound. Studies consistently show that female survivors report greater distress and anxiety about fertility than their male counterparts, often manifesting as social withdrawal, emotional turmoil, and depressive symptoms [66]. Parents may underestimate these concerns, focusing instead on survival milestones. Fertility-related distress typically intensifies with age, peaking around 10 years post diagnosis and adversely affecting romantic relationships and family planning [67].

Figure 3 outlines the key aspects of fertility outcomes in pediatric female cancer survivors, including the risks of premature ovarian insufficiency, methods for assessing ovarian reserve, pregnancy-related complications, and menstrual recovery rates.

### 4.2. Pediatric Male Patients

As discussed earlier, many oncologic treatments have gonadotoxic effects that can compromise male fertility—either temporarily or permanently. The primary mechanisms include impaired spermatogenesis, hypogonadism (either primary or secondary to hypothalamic or pituitary dysfunction), and structural or functional abnormalities of the genitourinary tract [68]. While the current section does not explore in detail the impact of central nervous system (CNS) malignancies or their treatments, it is important to note that the hypothalamic–pituitary–gonadal (HPG) axis can be disrupted by brain tumors, neurosurgery, cranial irradiation, and HSCT conditioning regimens [69].

In 2017, the International Late Effects of Childhood Cancer Guideline Harmonization Group (IGHG), in collaboration with the PanCareSurFup (PCSF) consortium, released evidence-based recommendations for the surveillance of reproductive health in male survivors of childhood cancer who underwent chemotherapy, radiotherapy, or surgery [70].

Unlike Leydig cells, which are relatively radioresistant, the germinal epithelium is highly susceptible to cytotoxic damage. Thus, impaired spermatogenesis may occur even in the absence of testosterone deficiency [69,71]. Spermatogenesis, the process of sperm cell production, occurs within the seminiferous tubules under the guidance of Sertoli cells and is regulated by follicle-stimulating hormone (FSH), luteinizing hormone (LH), and testosterone produced by Leydig cells [72].

Contrary to earlier beliefs, testicular activity is not quiescent during infancy and childhood. Sertoli and Leydig cells proliferate, and testicular volume increases during this phase [73]. Therefore, cytotoxic therapies administered during early developmental stages can have significant and lasting impacts on male reproductive capacity, irrespective of pubertal status at the time of treatment [74].

Impaired spermatogenesis encompasses a wide spectrum of conditions, including (1) azoospermia, the complete absence of sperm in the ejaculate; (2) oligozoospermia, reduced sperm count; (3) asthenozoospermia, reduced sperm motility; (4) teratozoospermia, abnormal sperm morphology [75].

Spermatogenic failure is commonly associated with high-dose chemotherapy (e.g., cyclophosphamide, chlormethine, procarbazine, ifosfamide, busulfan, fludarabine, and melphalan), HSCT conditioning regimens, and radiation that affects the testes [70,75]. Notably, even low radiation doses can cause acute spermatogenic damage, with doses ≥2–3 Gy associated with a high likelihood of permanent infertility [70].

Semen analysis remains the gold standard for evaluating spermatogenesis and should be performed upon patient request. An evaluation of Tanner stage, testicular volume, serum FSH, and inhibin B levels are also recommended for gonadal function monitoring [70]. In cases of severe oligozoospermia (sperm count ≤ 5 × 10^6^/mL), prior exposure to gonadotoxic treatments, or failed conception after 6 months of attempts, referral to a reproductive urologist is advised [70].

The recovery of spermatogenesis following cancer treatment is possible, as shown in some long-term follow-up studies involving adult and pediatric survivors [8]. However, the paucity of pediatric-specific data underscores the need for further research to better understand fertility recovery timelines and predictors [4,76,77].

Leydig cells, responsible for testosterone production, are more resistant to chemotherapy and radiation than germ cells [68,78]. Nonetheless, Leydig cell dysfunction (LCD) or failure (LCF) can occur, particularly following testicular radiation ≥12 Gy or total body irradiation (TBI) [70]. Testosterone deficiency may be transient or permanent and affects puberty onset in prepubertal males and sexual function and physical health in postpubertal individuals. Symptoms include a reduced libido, impaired secondary sexual characteristics, fatigue, and metabolic alterations such as insulin resistance and obesity [68,70,78].

The 2019 St. Jude Lifetime Cohort Study by Chemaitilly et al. confirmed testicular irradiation as a major risk factor for LCD and LCF. It also identified high CEDs (≥4000 mg/m^2^) as an independent risk factor for Leydig cell decline, with higher cumulative doses associated with progressive dysfunction over time [78].

Surveillance for Leydig cell function should include the assessment of Tanner staging, testicular volume, and serum testosterone and LH concentrations. In prepubertal and peripubertal males, growth velocity and pubertal progression should be monitored annually beginning at age 12. In postpubertal males, early morning serum testosterone should be evaluated, with LH measurement if levels are below the reference range [70]. Referral to endocrinology is warranted for cases of absent or arrested puberty (no signs by age 14) or established hypogonadism requiring hormone replacement therapy [68,70,71]. Testosterone therapy promotes normal pubertal development, enhances bone density and muscle mass, supports psychosocial well-being, and optimizes final height [71].

Sexual dysfunctions in male survivors can also result from structural damage—due to pelvic or spinal surgeries, radiotherapy involving the pelvic region, or injury to the sympathetic nervous system [70]. Although no universal screening recommendations exist for sexual dysfunction, clinicians should obtain a detailed sexual history, with referrals made in cases of positive findings [70].

Given the increased survival rates among pediatric cancer patients, addressing future reproductive potential is essential. A 2018 study revealed that 35.9% of childhood cancer survivors (CCSs) had inaccurate perceptions of their infertility risk based on their treatment history, leading to suboptimal reproductive decision-making and psychosocial distress [79]. In a Dutch cross-sectional study, although most male CCSs expressed a desire for fatherhood, 25% had not fulfilled it—compared with only 7% of their healthy siblings [80,81]. Furthermore, while male CCSs were three times more likely to consult a reproductive specialist (34% vs. 12%), their actual use of assisted reproductive technologies was markedly lower (41% vs. 77%) [82].

These findings emphasize the critical importance of integrating fertility preservation discussions at diagnosis. Collaboration between oncologists, endocrinologists, and fertility specialists is essential to ensure timely intervention, informed decision-making, and appropriate medical and psychological support for those affected by infertility.

Figure 4 illustrates the critical aspects of fertility in pediatric male cancer survivors, including variability in spermatogenesis recovery, the role of semen analysis in fertility assessment, testosterone deficiency as a long-term risk, and the absence of increased abnormalities in offspring.

### 4.3. From Survivorship to Parenthood: Clinical Challenges in the Transition Phase

Although the biological risks of infertility in pediatric cancer survivors are increasingly well characterized, the transition from medical survivorship to reproductive decision-making in adulthood remains poorly structured and inconsistently addressed. This phase—spanning adolescence, young adulthood, and reproductive planning—presents unique medical, psychological, and informational challenges that are often overlooked in standard follow-up protocols.

Many survivors report receiving limited or no counseling about long-term fertility implications at the time of treatment, and even fewer are provided with clear guidance as they transition into adulthood. This information gap becomes particularly problematic as survivors reach reproductive age and begin considering family building. In multiple studies, adult survivors expressed confusion or surprise upon learning, sometimes too late, that their ability to conceive had been compromised. This late awareness not only affects reproductive outcomes but also contributes to emotional distress, relationship difficulties, and altered life planning.

A major contributor to this is the lack of formalized transition models that ensure the continuity of care between pediatric oncology, adolescent medicine, endocrinology, and adult reproductive services. Fertility status is often not reassessed in follow-up visits unless the patient proactively raises the issue—something that many young adults are reluctant or unprepared to do. Moreover, routine survivorship care tends to focus on organ toxicity, second malignancies, or cardiovascular health, leaving reproductive health under-prioritized despite its long-term psychosocial relevance.

Additionally, even when fertility preservation was discussed or attempted during treatment, the outcomes of such interventions are rarely revisited later. For example, stored gametes or gonadal tissue may remain unused or unassessed for viability. In some cases, patients are unaware of how or where their biological material is stored, or they encounter unexpected logistical or financial barriers when they seek to use it. Prepubertal survivors, in particular, may have undergone experimental procedures such as testicular or ovarian tissue cryopreservation with uncertain long-term efficacy, and few systems are in place to follow these patients longitudinally or provide individualized counseling once reproductive decisions arise.

Another challenge lies in the variability of reproductive potential over time. Some survivors may experience partial or delayed gonadal recovery, while others develop premature gonadal failure many years after treatment. Without the regular assessment of hormone levels, semen analysis, ovarian reserve markers, and pubertal progression, changes in fertility status can go undetected until the survivor actively attempts conception.

From a systemic perspective, integrating fertility surveillance into survivorship care requires collaboration across disciplines and institutions. Oncologists, endocrinologists, gynecologists, urologists, reproductive specialists, psychologists, and primary care providers must be aligned around a shared framework that views fertility not as a static risk but as a dynamic, evolving aspect of survivorship. Age-appropriate fertility counseling should be reintroduced at critical milestones—puberty, completion of growth, and family planning stages—and should include both biological assessments and psychosocial support.

Ultimately, the goal is to move beyond a reactive model—where fertility discussions occur only when survivors report problems—and toward a proactive, anticipatory care approach. Creating individualized survivorship plans that include reproductive health trajectories, offering fertility “check-ins” during adolescence and young adulthood, and ensuring accessible pathways to assisted reproductive technologies (ARTs) are essential steps. This paradigm shift can help survivors reclaim agency over their reproductive futures and facilitate the transition from survival to parenthood as a realistic, supported outcome of modern cancer care.

Figure 5 highlights the main clinical challenges faced by childhood cancer survivors during the transition from survivorship to parenthood, including reproductive desires, medical complexity, contraception needs, and reliance on assisted reproductive technologies.

## 5. Psychosocial Impact of Fertility Loss in Childhood Cancer Survivors

Fertility loss is not only a biological consequence of cancer treatment but also a significant psychosocial burden that can affect survivors throughout their lives. Multiple studies highlight that fertility-related distress ranks among the most critical concerns reported by childhood cancer survivors, particularly females, and may contribute substantially to a reduced quality of life [66,67].

A large systematic review analyzing 27 studies found that 30–75% of female survivors expressed moderate to high levels of concern about infertility and its impact on their future relationships and family planning [83]. Women survivors are consistently more distressed by fertility issues than male survivors, with some studies reporting anxiety and depressive symptoms nearly double those observed in male counterparts [66]. In a cross-sectional analysis by Lehmann et al., female survivors reported significantly higher levels of emotional turmoil, social withdrawal, and diminished self-esteem when grappling with fertility uncertainty [67].

The psychosocial burden often evolves with age. During treatment and immediately post therapy, fertility may not be perceived as a priority due to the overwhelming focus on survival. However, concerns tend to intensify over time. A recent analysis of over 3000 survivors revealed that the prevalence of fertility-related distress rises sharply in survivors entering young adulthood, peaking between 10 and 15 years post treatment, coinciding with typical life milestones such as forming intimate relationships and contemplating parenthood [66]. Notably, survivors aged 25–35 reported the highest rates of reproductive anxiety and dissatisfaction with information received during cancer treatment [84].

Beyond individual distress, fertility concerns have substantial implications for social and family dynamics. Communication barriers between survivors and their families are common. Many parents underestimate the significance of fertility for their children or avoid discussing it due to discomfort or cultural taboos [66]. Conversely, adolescents and young adults may feel inhibited from initiating such conversations, fearing embarrassment or the belief that discussing fertility could seem trivial in the context of a life-threatening illness. This silence may foster misconceptions, amplify distress, and prevent timely engagement with fertility preservation services.

Family planning decisions are further complicated by uncertainties about fertility status and the potential risks of genetic transmission of cancer predispositions, adding layers of anxiety for survivors considering parenthood [67]. Studies show that women survivors often experience significant ambivalence towards pregnancy, torn between their desire for biological children and fear of complications or adverse offspring outcomes due to prior treatments [83].

Psycho-oncology services play an essential role in mitigating these challenges. Psycho-oncologists and reproductive psychologists can provide survivors and their families with accurate information, facilitate open communication, and help manage the psychological consequences of fertility loss. Interventions such as cognitive behavioral therapy (CBT), fertility-focused counseling, and peer support groups have shown efficacy in reducing fertility-related distress and improving quality of life [83,84].

Despite these advances, significant gaps remain in integrating psychosocial care into fertility preservation pathways. Studies have reported that only 30–50% of survivors recall receiving fertility information at diagnosis, and fewer still report access to psychological support services tailored to reproductive health concerns [66,67]. This underscores the need for a multidisciplinary approach in pediatric oncology, ensuring that fertility preservation discussions include not only the medical risks and technical options but also the profound emotional and social implications of potential infertility.

In conclusion, fertility preservation should be framed not solely as a medical intervention but as an integral component of holistic survivorship care. Early, age-appropriate counseling, ongoing psychological support, and the proactive involvement of psycho-oncology professionals are crucial to alleviating the long-term psychosocial burden of fertility loss among childhood cancer survivors. From a psychosocial standpoint, one of the most underexplored areas remains the longitudinal trajectory of fertility-related distress in survivors who reach adulthood without prior preservation. These individuals often face profound uncertainty—unable to confirm their fertility status but reluctant to undergo invasive testing or costly ART consultations. This ambiguous fertility state contributes to emotional paralysis and delays in family planning decisions. Integrating structured reproductive health assessments into young adult survivorship care could mitigate this uncertainty and empower survivors with actionable information. Additionally, tailored psychosocial interventions for this population—distinct from those offered at diagnosis—are needed to address the evolving emotional landscape of fertility concerns in adult life.

Figure 6 illustrates the psychosocial consequences of fertility loss in childhood cancer survivors, encompassing emotional distress, identity challenges, relationship difficulties, and perceived social stigma.

## 6. Future Perspectives

In considering the future of fertility care for pediatric cancer survivors, it is imperative to move beyond purely technical preservation strategies and examine the broader continuum of reproductive health. Survivors’ needs evolve over time, and fertility preservation should be viewed not as a single intervention, but as the starting point of a dynamic, lifelong process that includes surveillance, counseling, and eventual access to assisted reproduction when necessary. Furthermore, as the field advances, ethical questions surrounding experimental techniques in prepubertal children, consent capacity, and equity of access must be re-evaluated in light of new data and societal expectations. These broader considerations are essential to ensure that fertility care keeps pace with both medical progress and survivor aspirations.

Table 1 shows the principal effects of cancer treatments (chemotherapy, radiotherapy, and the combination of both) on fertility in both males and females, whereas Table 2 summarizes the different fertility preservation techniques in both prepubertal and postpubertal males and females.

As survivorship rates continue to improve, the long-term efficacy of fertility preservation treatments remains a critical area of research in pediatric oncology. While many patients recover gonadal function after chemotherapy or radiotherapy, predicting individual outcomes is still imprecise. Longitudinal studies assessing the durability of ovarian tissue grafts and the recovery of spermatogenesis after testicular tissue transplantation are urgently needed to establish robust efficacy data and refine risk stratification models. In parallel, the psychosocial outcomes following fertility interventions require greater exploration. Although fertility preservation is associated with reduced long-term distress, the emotional burden of invasive procedures, treatment delays, and lingering uncertainty can adversely affect mental health and quality of life, particularly among young female survivors [66,67,83,84]. Future research should include not only reproductive endpoints but also comprehensive psychosocial assessments to evaluate survivors’ satisfaction, relational outcomes, and psychosocial resilience after fertility preservation.

Additionally, the expanding use of novel targeted therapies introduces new uncertainties regarding gonadotoxicity. Agents such as tyrosine kinase inhibitors (TKIs) and immunotherapies—including CAR-T cells—have revolutionized pediatric cancer care, but data on their reproductive risks remain scarce. Some TKIs may disrupt hormonal pathways, while CAR-T therapy, although promising in treating refractory malignancies, is associated with cytokine release syndromes that could indirectly affect gonadal function. The potential for these agents to cause direct or indirect gonadotoxicity underscores the importance of prospective fertility monitoring and basic research into their reproductive impact.

Addressing these knowledge gaps is essential to optimize patient counseling and clinical practice. Table 3 summarizes the key unmet needs and research priorities in pediatric fertility preservation.

## 7. Conclusions

With the notable rise in survival rates among pediatric cancer patients over the past few decades, fertility preservation and long-term reproductive health have become critical components of survivorship care.

The gonadotoxic effects of cancer treatments—including chemotherapy, radiotherapy, and surgery—are now well recognized and their impact on reproductive function is closely tied to cumulative doses, treatment timing, and the patient’s age at exposure. As such, fertility counseling should be an integral part of the initial oncology consultation. Discussing the risks and preservation strategies early allows families to make informed decisions and supports better psychological adjustment throughout treatment and recovery.

While standardized fertility preservation techniques are now available for postpubertal patients of both sexes, options for prepubertal children remain largely experimental and ethically complex. Multidisciplinary collaboration is essential—not only to guide the choice of fertility-preserving interventions but also to ensure a comprehensive follow-up plan that includes hormonal monitoring, the assessment of pubertal progression, and the evaluation of reproductive function.

Despite progress, significant knowledge gaps persist, particularly in the pediatric population. Longitudinal studies assessing the long-term efficacy, safety, and psychosocial outcomes of fertility preservation strategies in children are urgently needed. In female patients, further research is required to clarify the impact of novel targeted therapies and to improve protocols for ovarian tissue transplantation. In males, advances in spermatogonial stem cell research and testicular tissue transplantation offer promising avenues, but clinical translation remains limited. Additionally, prospective, age-stratified studies evaluating the recovery of gonadal function post treatment would help establish more precise risk stratification and inform follow-up protocols.

Equally important is the need for research that addresses the disparities in access to fertility care, particularly for patients from socioeconomically disadvantaged backgrounds. Future work should also examine how fertility concerns evolve over time and affect identity, relationships, and the quality of life in cancer survivors. A patient-centered approach that integrates medical, psychological, and social dimensions is essential to improving long-term outcomes and supporting the transition from survival to parenthood.

## Figures and Tables

**Figure 1 biomedicines-13-01859-f001:**
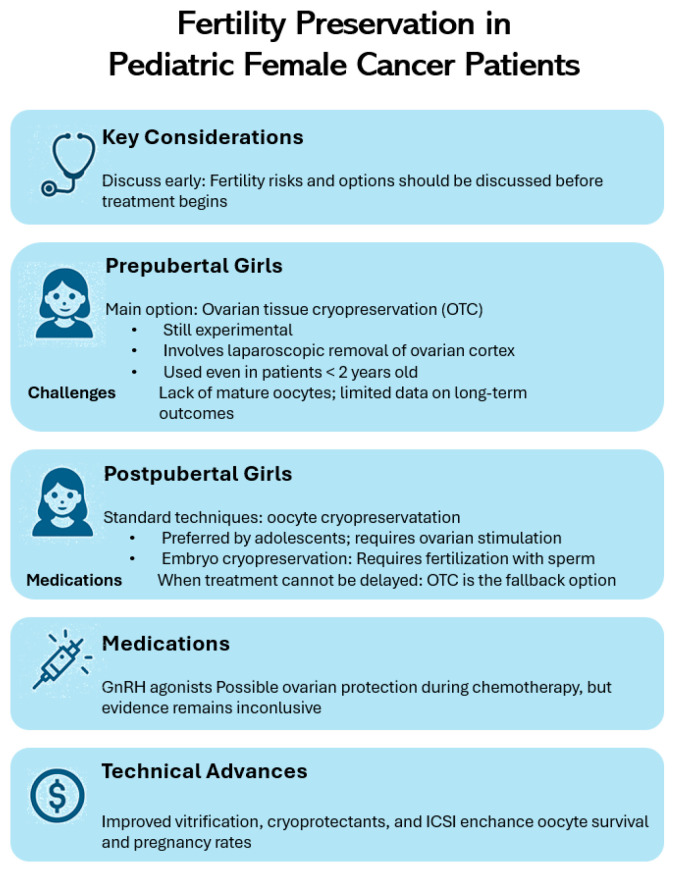
Fertility preservation options and considerations for pediatric female cancer patients.

**Figure 2 biomedicines-13-01859-f002:**
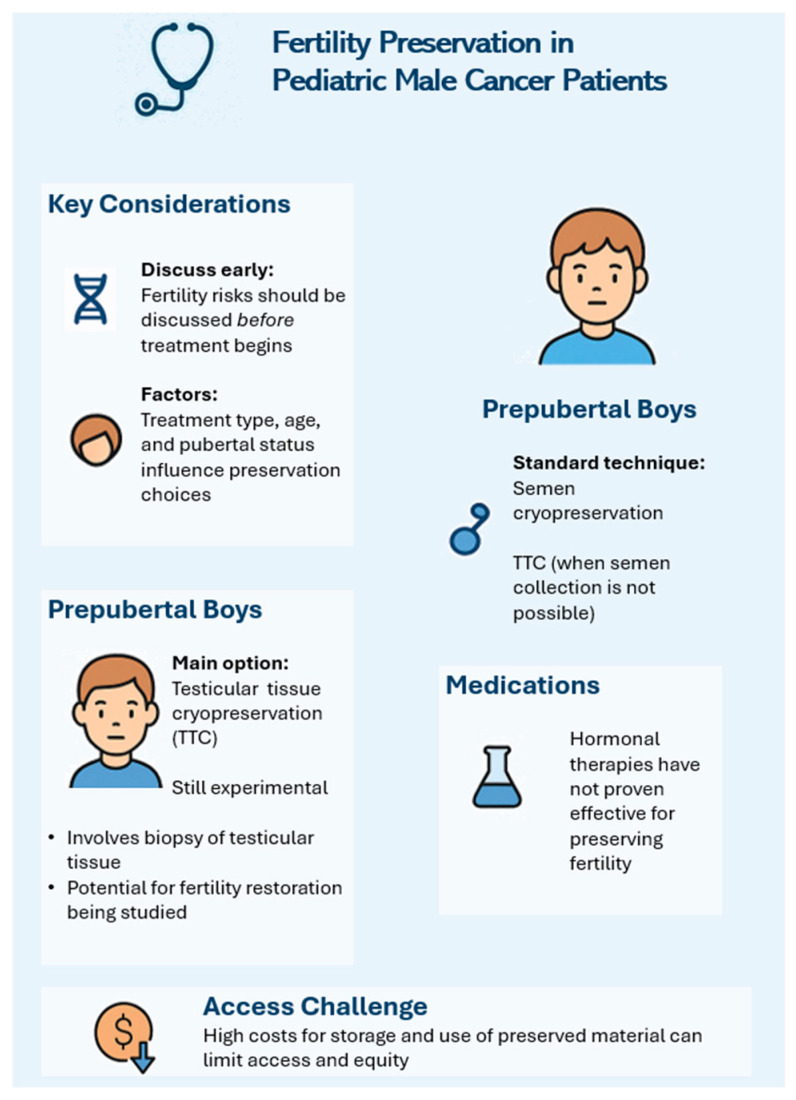
Fertility preservation approaches in pediatric male cancer patients.

**Figure 3 biomedicines-13-01859-f003:**
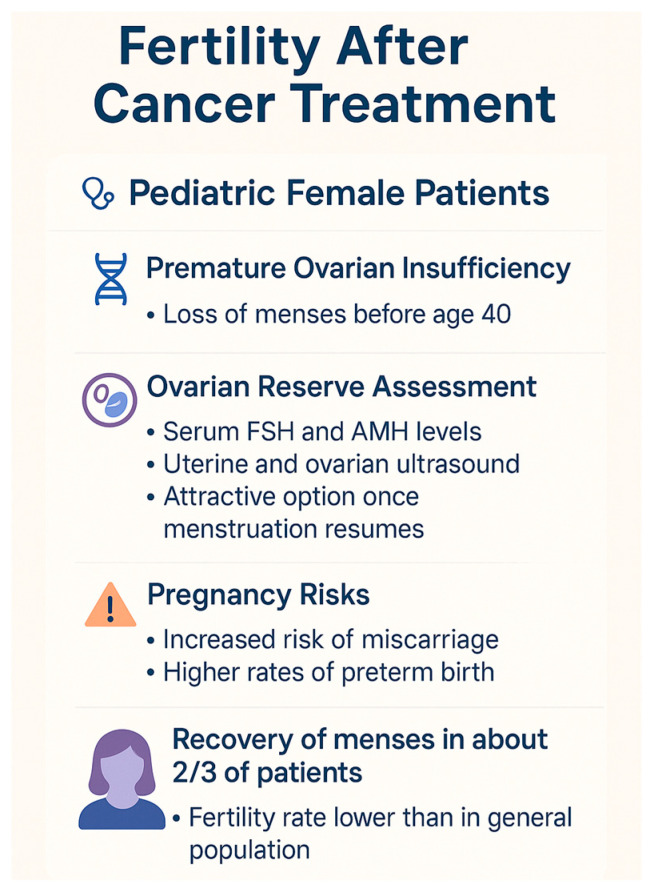
Key aspects of fertility outcomes in pediatric female cancer survivors.

**Figure 4 biomedicines-13-01859-f004:**
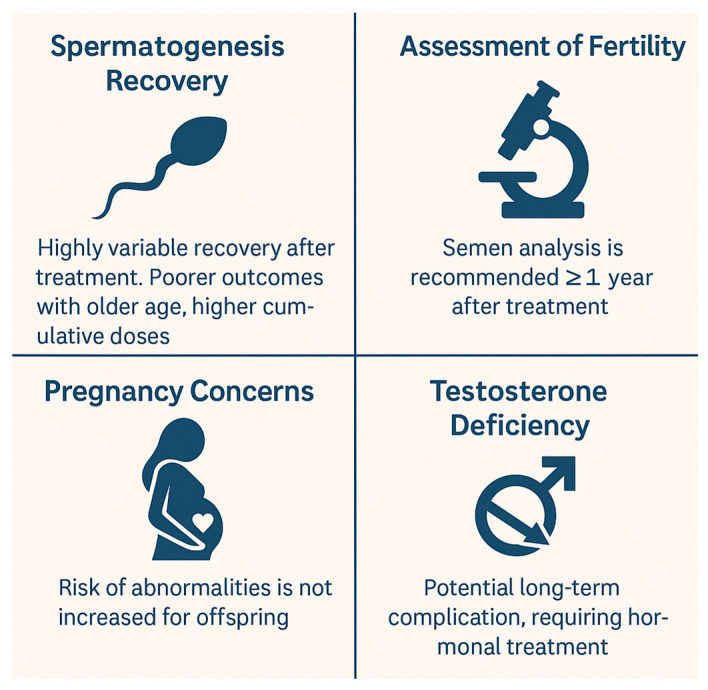
Critical aspects of fertility in pediatric male cancer survivors.

**Figure 5 biomedicines-13-01859-f005:**
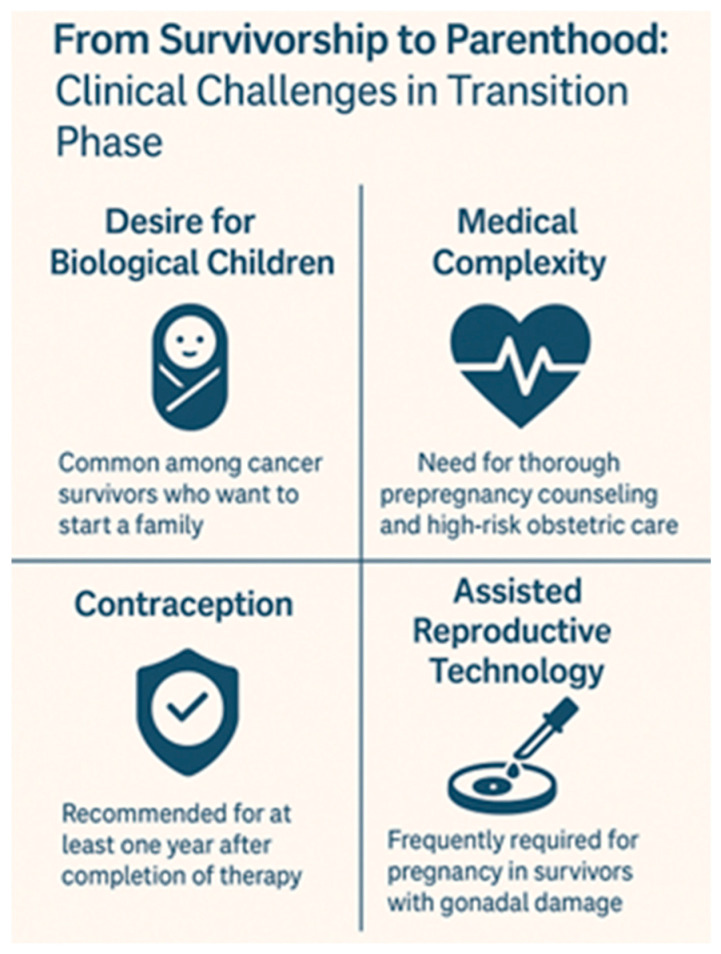
Main clinical challenges faced by childhood cancer survivors during the transition from survivorship to parenthood.

**Figure 6 biomedicines-13-01859-f006:**
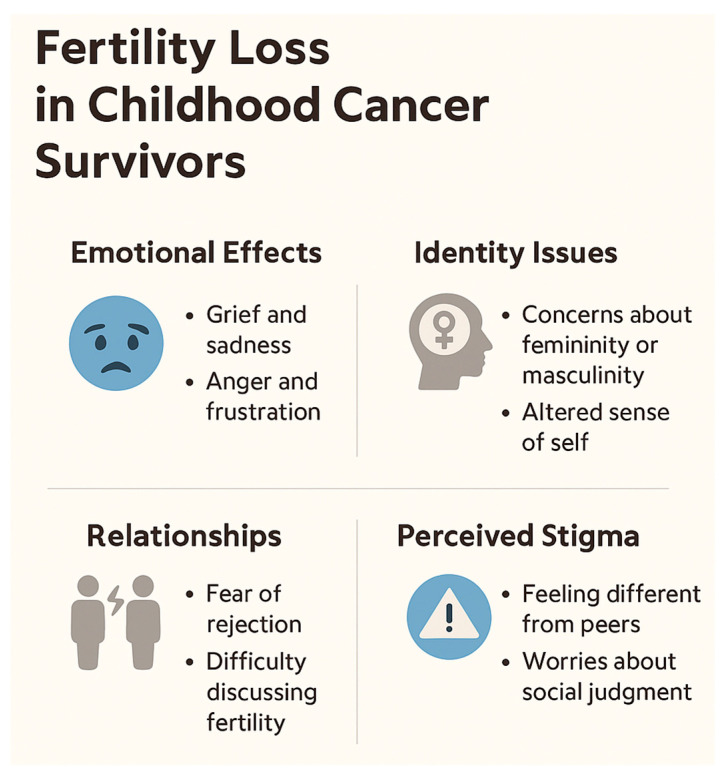
Psychosocial consequences of fertility loss in childhood cancer survivors.

**Table 1 biomedicines-13-01859-t001:** The principal effects of cancer treatments (chemotherapy, radiotherapy, and the combination of both) on fertility in both males and females.

Type of Therapy	Effects on Females	Effects on Males
**Alkylating agents**	High risk of premature ovarian insufficiency (POI) [57,60]	Risk of azoospermia or oligospermia [70]
**Non-alkylating agents**	Moderate or low gonadotoxicity (e.g., taxanes, anthracyclines) [59]	Generally reduced gonadotoxic effects (e.g., vincristine, methotrexate) [70]
**Radiotherapy directed on the gonads**	Direct ovarian damage, with POI risk even at low doses (>1 Gy) [43,46]	Azoospermia almost certain with doses >2–3 Gy [70]
**Total Body Irradiation (TBI)**	Nearly certain ovarian insufficiency with high doses (>4 Gy) [70]	Damage to germ cells and Leydig cells with exposure >12 Gy [70]
**Stem Cell Transplantation**	Very high impact on fertility, especially if associated with preparatory regimens based on high doses of cyclophosphamide or busulfan [61]	Severe impact on spermatogenesis and Leydig cell function [78]

**Table 2 biomedicines-13-01859-t002:** Different fertility preservation techniques in both prepubertal and postpubertal males and females.

Technique	Gender	Target Age	Current Status	Notes
**Ovarian Tissue Cryopreservation**	Female	Prepubertal and postpubertal	Experimental	- Only option for prepubertal girls - Requires laparoscopy for tissue collection [18,20]
**Oocyte Cryopreservation**	Female	Postpubertal	Standardized	- Preferred by young patients without fertilizing embryos - Requires ovarian stimulation [18,22]
**Embryo Cryopreservation**	Female	Postpubertal	Standardized	- Requires fertilized oocytes - Less ethically problematic [18,23]
**Testicular Tissue Cryopreservation**	Male	Prepubertal	Experimental	- Potential future use with spermatogonia stem cells or transplants [12,32]
**Sperm Cryopreservation**	Male	Postpubertal	Standardized	- Well-established technique - Requires collection via self-stimulation [12,27]

**Table 3 biomedicines-13-01859-t003:** Unmet needs and research priorities in pediatric fertility preservation.

Domain	Unmet Needs	Research Priorities
Long-term efficacy of preservation	Limited data on live birth rates post cryopreservation; uncertainty about graft longevity	Prospective studies on ovarian tissue transplantation outcomes; monitoring of spermatogenic recovery post TTC
Psychosocial impact	Insufficient integration of psychosocial care into fertility pathways; gaps in counseling	Studies on survivor satisfaction and mental health after fertility preservation; development of tailored interventions
Impact of novel therapies	Sparse data on fertility effects of TKIs, CAR-T, immunotherapies	Preclinical and clinical studies to determine gonadotoxicity profiles; fertility monitoring guidelines for new drugs
Access and equity	Socioeconomic disparities in fertility care	Policies to improve funding and universal access to preservation services

## Data Availability

Not applicable.

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
