# Peer review of "From Survival to Parenthood: The Fertility Journey After Childhood Cancer"

_biomedicines, 2025, doi:10.3390/biomedicines13081859_

Round 1

Reviewer 1 Report

Comments and Suggestions for Authors

They have put together their findings on the subject area; few major comments need to be addressed

Comments

I have seen a similar paper online, :Spix C. Fertility in survivors of childhood cancer. Dtsch Arztebl Int. 2012 Feb;109(7):124-5. doi: 10.3238/arztebl.2012.0124. Epub 2012 Feb 17. PMID: 22427789; PMCID: PMC3301973.". How would you justify the novelty of your paper. 

It is very important that such a clarification or justification is given since, as it stands, the current similarity stands unjustified. 

Also, the authors have done a meta analysis sort, some statistics and figures shud be provided. the data profile of the paper is insufficient

The last para of Introduction usually highlights what has been donein the paper and the significance and novelty of the proposed work. Nothin of these have been reported. 

The paper appears to have been very hastily put together , with format and font size variation evident throughout.

The depth of the study and the analysis is un justified

Will require a thorough revision

Comments on the Quality of English Language

Major revise

Author Response

I have seen a similar paper online, :Spix C. Fertility in survivors of childhood cancer. Dtsch Arztebl Int. 2012 Feb;109(7):124-5. doi: 10.3238/arztebl.2012.0124. Epub 2012 Feb 17. PMID: 22427789; PMCID: PMC3301973.". How would you justify the novelty of your paper. It is very important that such a clarification or justification is given since, as it stands, the current similarity stands unjustified. 

Re: We clarified at the end of the Introduction (p. 2) that although numerous reviews have addressed fertility issues in childhood cancer survivors, significant gaps remain regarding sex-specific risks, experimental options for prepubertal patients, and the psychosocial consequences of fertility loss. Moreover, recent advances in reproductive technologies and updated clinical guidelines necessi-ate a comprehensive synthesis of current evidence. This review aims to fill this gap by providing an up-to-date, detailed, and practical resource for clinicians managing pediatric patients facing fertility risks.

Also, the authors have done a meta analysis sort, some statistics and figures shud be provided. the data profile of the paper is insufficient.

Re: It is clearly written that this is a narrative review and details for study evaluation have been reported.

The last para of Introduction usually highlights what has been donein the paper and the significance and novelty of the proposed work. Nothin of these have been reported. 

Re: Revised accordingly (p. 2).

The paper appears to have been very hastily put together , with format and font size variation evident throughout.

The depth of the study and the analysis is un justified

Will require a thorough revision

Re: Thank you for your suggestions. The manuscript has been revised according to your comments and those received from the other reviewers.

Reviewer 2 Report

Comments and Suggestions for Authors

Although the authors try to write a comprehensive review entitled 'From Survival to Parenthood: The Fertility Journey After Childhood Cancer,' they fail in their objective. The first question is why the authors chose this topic when a large number of similar, already published articles are available. There is a serious question and concern regarding the novelty of the work, and the author fails to justify the reasons behind the same. Secondly, the article is poorly organized. The introduction is just a summary of infertility without a clear explanation of the objective of the work. The entire work suffers from several grammatical and typographical errors. The font size is not uniform throughout the manuscript, which makes it difficult to follow. There are six main headings and no proper subheadings within these, which clearly shows the lack of depth of this work. This work simply looks like a summarized version of previously published articles. The lack of self-explanatory scientific diagrams further reduces the quality of the work. 

Comments on the Quality of English Language

The English should be improved, as it lacks a scientific touch

Author Response

Although the authors try to write a comprehensive review entitled 'From Survival to Parenthood: The Fertility Journey After Childhood Cancer,' they fail in their objective. The first question is why the authors chose this topic when a large number of similar, already published articles are available. There is a serious question and concern regarding the novelty of the work, and the author fails to justify the reasons behind the same.

Re: We clarified at the end of the Introduction (p. 2) that although numerous reviews have addressed fertility issues in childhood cancer survivors, significant gaps remain regarding sex-specific risks, experimental options for prepubertal patients, and the psychosocial consequences of fertility loss. Moreover, recent advances in reproductive technologies and updated clinical guidelines necessitate a comprehensive synthesis of current evidence. This review aims to fill this gap by providing an up-to-date, detailed, and practical resource for clinicians managing pediatric patients facing fertility risks.

Secondly, the article is poorly organized. The introduction is just a summary of infertility without a clear explanation of the objective of the work. The entire work suffers from several grammatical and typographical errors. The font size is not uniform throughout the manuscript, which makes it difficult to follow. There are six main headings and no proper subheadings within these, which clearly shows the lack of depth of this work.

Re: We improved the organization of the manuscript. We followed Instruction to authors of the journal. Moreover, the text has been revised by an English mother tongue with appropriate knowledge of the topic.

This work simply looks like a summarized version of previously published articles. The lack of self-explanatory scientific diagrams further reduces the quality of the work.

Re: Further details have been added in the Methods (pp. 3-4). However, the PRISMA diagram has not been added because this is a narrative review. Moreover, several sections have been improved according to the suggestions received from the reviewers.

Reviewer 3 Report

Comments and Suggestions for Authors

Dear authors

We thank you for submission of your manuscript to Biomedicines. This review touches upon an ever-relevant subject—the impacts of pediatric cancer therapy on fertility and the state of preservation methods. Your manuscript reads easily with sufficient material. However, there are essential methodological clarifications, structural adjustments, and literature updates prior to our acceptance of the manuscript for publishing. Review the following remarks for thorough revisions of significance:

1. Explain Review Approach (Section 2): Although it's a narrative review, the methods section needs to be more transparent. Please say:

The number of articles screened and selected;

Whether an additional PRISMA-style diagram can be provided

If any quality assessment tool for study eligibility was applied (e.g., use of the RoB tools).

2. Balance Between the Content of Both Genders: Even though the female part of fertility is comprehensive, the male side is quite compact. Richen it with:

Building upon the future clinical potential and long-term prospects of testicular tissue cryopreservation;

Identifying the procedural as well as the psychological barriers to sperm procurement among adolescent boys;

Providing additional information regarding SSC and in vitro spermatogenesis studies.

3. Terminological Consistency: Clearly differentiate and describe "acute ovarian failure," "premature ovarian insufficiency," and "premature menopause" with the assistance of authority sources like the ESHRE 2016

4. Closing Research Gaps: Even though areas where more research is needed are mentioned in the conclusions, the body of the manuscript does not have direct discussion about:

Long-term efficacy of treatments for recovery of fertility

Psychosocial function following intervention;

The impact of novel therapies (e.g., CAR-T, TKIs) upon fertility. Clarity would be provided by an organizational summary table of "Unmet Needs and Research Priorities."

5. Section 5 and Section 6: Psychosocial Impact. This section is too brief given how clinically relevant it is.

Elaborate with evidence-based data about anxiety with respect to fertility, especially in women survivors.

The age-related progression of such problems;

Discuss family relationships, communication deficits, and the role of psycho-oncology.

6. Include Current Key References: You omit two critical papers that are very relevant to the domain of the manuscript:

https://doi.org/10.3390/cancers13246331

https://doi.org/10.1371/journal.pone.0308827

7. Edit Figures and Tables:

Add bibliographic references to Table 1 and Table 2.

Improve visual design with color-coding or highlight risk categories.

Consider incorporating a flowchart or decision tree to guide decision-making for fertility preservation based upon sex and pubertal status.

8. Increase Structural Clarity: Certain of the paragraphs (particularly Sections 3–6) are excessively long and have several topics. Steer clear of duplicating ideas (e.g., tissue cryopreservation of the ovaries has little variation in many cases).

9. Update and Validate Guidelines: Some references to certain guidelines are outdated (e.g., ASCO 2013). Look for:

ASCO Fertility Preservation Update 2018

ESHRE POI guideline 2022

Other pediatric oncofertility position statements, like ESE, SIOP

This is a good and thought-provoking manuscript that would make an excellent source of reference for pediatric oncologists, endocrinologists, and reproductive experts. However, the manuscript needs critical improvements in methodology description, balance in terms of gender, and synthesis of literature. Resubmission after intensive overhaul with due care for all the above mentioned points is recommended.

Author Response

Dear authors

We thank you for submission of your manuscript to Biomedicines. This review touches upon an ever-relevant subject—the impacts of pediatric cancer therapy on fertility and the state of preservation methods. Your manuscript reads easily with sufficient material. However, there are essential methodological clarifications, structural adjustments, and literature updates prior to our acceptance of the manuscript for publishing. Review the following remarks for thorough revisions of significance:

Re: Thank you for your comments. We revised the manuscript according to your suggestions and those received from the other reviewers.

  1. Explain Review Approach (Section 2): Although it's a narrative review, the methods section needs to be more transparent. Please say:

The number of articles screened and selected;

Whether an additional PRISMA-style diagram can be provided

If any quality assessment tool for study eligibility was applied (e.g., use of the RoB tools).

Re: Further details have been added (pp. 3-4). However, the PRISMA diagram has not been added because this is a narrative review.

  1. Balance Between the Content of Both Genders: Even though the female part of fertility is comprehensive, the male side is quite compact. Richen it with:

Building upon the future clinical potential and long-term prospects of testicular tissue cryopreservation;

Identifying the procedural as well as the psychological barriers to sperm procurement among adolescent boys;

Providing additional information regarding SSC and in vitro spermatogenesis studies.

Re: the text has been improved according to your comments (pp. 4-6).

  1. Terminological Consistency: Clearly differentiate and describe "acute ovarian failure," "premature ovarian insufficiency," and "premature menopause" with the assistance of authority sources like the ESHRE 2016

Re: Done as suggested (p. 6).

  1. Closing Research Gaps: Even though areas where more research is needed are mentioned in the conclusions, the body of the manuscript does not have direct discussion about:

Long-term efficacy of treatments for recovery of fertility

Psychosocial function following intervention;

The impact of novel therapies (e.g., CAR-T, TKIs) upon fertility. Clarity would be provided by an organizational summary table of "Unmet Needs and Research Priorities."

Re: A specific section on Future perspective has been added (pp. 11-13).

  1. Section 5 and Section 6: Psychosocial Impact. This section is too brief given how clinically relevant it is.

Elaborate with evidence-based data about anxiety with respect to fertility, especially in women survivors.

The age-related progression of such problems;

Discuss family relationships, communication deficits, and the role of psycho-oncology.

Re: A specific section has been added (pp. 10-11).

  1. Include Current Key References:

Re: Done as suggested (p. 20).

  1. Edit Figures and Tables:

Add bibliographic references to Table 1 and Table 2.

Improve visual design with color-coding or highlight risk categories.

Consider incorporating a flowchart or decision tree to guide decision-making for fertility preservation based upon sex and pubertal status.

Re: Tables 1 and 2 have been improved and references are included. Moreover, a new Table has been added.

  1. Increase Structural Clarity: Certain of the paragraphs (particularly Sections 3–6) are excessively long and have several topics. Steer clear of duplicating ideas (e.g., tissue cryopreservation of the ovaries has little variation in many cases).

Re: Done as requested.

  1. Update and Validate Guidelines: Some references to certain guidelines are outdated (e.g., ASCO 2013). Look for:

ASCO Fertility Preservation Update 2018

ESHRE POI guideline 2022

Other pediatric oncofertility position statements, like ESE, SIOP

Re: Done as requested (pp. 15, 17 and 20).

 This is a good and thought-provoking manuscript that would make an excellent source of reference for pediatric oncologists, endocrinologists, and reproductive experts. However, the manuscript needs critical improvements in methodology description, balance in terms of gender, and synthesis of literature. Resubmission after intensive overhaul with due care for all the above mentioned points is recommended.

Re: Thank you very much for your suggestions. We revised the manuscript according to your comments.

Round 2

Reviewer 1 Report

Comments and Suggestions for Authors

Accept

Author Response

Thank you for the positive evaluation of our manuscript. We revised it according to the suggestions received from other reviewers. 

Reviewer 2 Report

Comments and Suggestions for Authors

Although the authors claim to have improved the manuscript, it still appears speculative without any concrete explanations. The authors need to refer to several recent articles to prepare a comprehensive review article. The authors have just divided the previous headings into subheadings, which does not make much sense. A more detailed and insightful explanation is needed for childhood cancers and the transition in the fertility journey thereafter. The sections require more diverse perspectives from the authors, rather than simply paraphrasing already published knowledge. Overall, I feel the authors need to critically evaluate each section, refer to more articles, and provide a more detailed explanation for the same.

Author Response

We sincerely thank the reviewer for the additional feedback and the opportunity to further strengthen our manuscript.

As previously noted, we had already conducted an extensive revision of the manuscript following the initial review round. We carefully restructured and expanded several sections—particularly those addressing the gonadotoxic impact of treatments, sex-specific preservation strategies, and psychosocial implications—with the aim of improving clarity, depth, and clinical utility.

We are pleased to report that two of the three reviewers expressed their full support for the revised manuscript and found it acceptable for publication. Nonetheless, we fully acknowledge the value of Reviewer #3’s remaining concerns, particularly regarding the need for a more detailed and original discussion on the transition from childhood cancer survivorship to parenthood.

In response, we have now added a new, expanded section titled “4.3. From Survivorship to Parenthood: Clinical Challenges in the Transition Phase”, which delves into the practical, organizational, and emotional challenges faced by survivors in adulthood. This includes discussion of long-term follow-up gaps, reproductive uncertainty, limitations of current fertility preservation models, and the need for integrated transition care between pediatric oncology and adult reproductive medicine. We have also included the authors’ critical reflections and proposed frameworks for multidisciplinary coordination, in line with the reviewer’s request for more diverse perspectives.

We hope that these additional revisions address your comments satisfactorily and that the manuscript may now meet your approval.

Reviewer 3 Report

Comments and Suggestions for Authors

Authors have significantly improved the manuscript. Now I approve that this study scientifically sounds so that is at a level of being acceptance for publication depending on the final decision of the editor in chief of the journal.

Bests

Author Response

Thank you very much for the acceptance of the revised version of our manuscript. We further revised it according to comments received from one of the reviewers.

Round 3

Reviewer 2 Report

Comments and Suggestions for Authors

The authors have tried to improve the manuscript. I believe that since this is a review article, detailed and mechanistic pictorial representations under each heading are needed to understand the topics better for a reader. Most of the issues have been addressed. The English language and grammatical errors can be improved a bit.

Author Response

Thank you for your comments. We added 6 Figures to the 3 Tables. Moreover, the text has been revised by a native English speaker with appropriate knowledge of the topic. We hope that you could accept our manuscript in its current version.